# Magnon bound states versus anyonic Majorana excitations in the Kitaev honeycomb magnet $\alpha$-RuCl$_3$

Dirk Wulferding[1,2,3,7 ✉], Youngsu Choi[4,7], Seung-Hwan Do[4], Chan Hyeon Lee[4], Peter Lemmens[1,2], Clément Faugeras [5], Yann Gallais [6] & Kwang-Yong Choi [4 ✉]

The pure Kitaev honeycomb model harbors a quantum spin liquid in zero magnetic fields, while applying finite magnetic fields induces a topological spin liquid with non-Abelian anyonic excitations. This latter phase has been much sought after in Kitaev candidate materials, such as $\alpha$-RuCl$_3$. Currently, two competing scenarios exist for the intermediate field phase of this compound ($B = 7 - 10$ T), based on experimental as well as theoretical results: (i) conventional multiparticle magnetic excitations of integer quantum number vs. (ii) Majorana fermionic excitations of possibly non-Abelian nature with a fractional quantum number. To discriminate between these scenarios a detailed investigation of excitations over a wide field-temperature phase diagram is essential. Here, we present Raman spectroscopic data revealing low-energy quasiparticles emerging out of a continuum of fractionalized excitations at intermediate fields, which are contrasted by conventional spin-wave excitations. The temperature evolution of these quasiparticles suggests the formation of bound states out of fractionalized excitations.

[1] Institute for Condensed Matter Physics, TU Braunschweig, D-38106 Braunschweig, Germany. [2] Laboratory for Emerging Nanometrology (LENA), TU Braunschweig, D-38106 Braunschweig, Germany. [3] Center for Correlated Electron Systems, Institute for Basic Science (IBS), Seoul 151-742, Republic of Korea. [4] Department of Physics, Chung-Ang University, Seoul 156-756, Republic of Korea. [5] University of Grenoble Alpes, INSA Toulouse, University of Toulouse Paul Sabatier, EMFL, CNRS, LNCMI, 38000 Grenoble, France. [6] Laboratoire Matériaux et Phénomènes Quantiques (UMR 7162 CNRS), Université de Paris, 75205 Paris Cedex 13, France. [7] These authors contributed equally: Dirk Wulferding, Youngsu Choi. ✉email: dirk.wulferding@tu-bs.de; kchoi@cau.ac.kr

The search for Majorana fermions in solid-state systems has led to the discovery of several promising candidate materials for exchange-frustrated Kitaev quantum spin systems[1–6]. One of the closest realizations of a Kitaev honeycomb lattice is $\alpha$-RuCl$_3$[7,8], where the spin Hamiltonian is dominated by Kitaev interaction $K$. Nevertheless, non-Kitaev interactions, such as Heisenberg ($J$) and off-diagonal symmetric exchange terms ("$\Gamma$-term"), as well as stacking faults in $\alpha$-RuCl$_3$ lead to an antiferromagnetically ordered zigzag ground state below $T_N \approx 7$ K[8]. The exact strengths of these interactions have not been pinpointed, yet a general consensus on the minimal model has emerged about a ferromagnetic $K \sim -6$ to $-16$ meV, as well as $\Gamma \sim 1$ to 7 meV, and $J \sim -1$ to $-2$ meV[9,10]. Despite these additional magnetic parameters, the Majorana fermionic quasiparticles are well preserved at high energies and elevated temperatures[11]. Indeed, many independent and complementary experimental techniques have been used to probe the emergence of itinerant Majorana fermions and localized gauge fluxes from the fractionalization of spin degrees of freedom[9,12,13].

A promising route in understanding Kitaev physics might be the suppression of long-range magnetic order in a magnetic field, with the possibility of generating an Ising topological quantum spin liquid. For $\alpha$-RuCl$_3$ this field is $B_c \sim 6$–7 T[14–16]. Higher magnetic fields lead to a trivial spin polarized state. In the intermediate field range of 7–10 T, the magnetic order melts into

a quantum disordered phase, in which the half-integer quantized thermal Hall conductance is reported[13]. This remarkable finding may be taken as evidence for a field-induced topological spin liquid with chiral Majorana edge states and the central charge $q = \nu/2$ (Chern number $\nu = 1$). However, it is less clear whether such a chiral spin liquid state can be stabilized in the presence of a relatively large field and non-Kitaev terms in $\alpha$-RuCl$_3$. In the original Kitaev honeycomb model, non-Abelian Majorana excitations are created upon breaking time-reversal symmetry, e.g., through applying a magnetic field[17]. These composite quasiparticles correspond to bound states of localized fluxes and itinerant Majorana fermions[18]. The composite bound states are of neither fermionic nor bosonic character, but instead they acquire an additional phase in the wavefunction upon interchanging particles, i.e., they follow anyonic statistics[19]. Although for a Kitaev system bound itinerant Majorana fermions can be formed in the presence of perturbations[18], it is unclear how stable a topological spin liquid state is in this case. In particular, an open issue is whether the quantized thermal Hall effect is related to a non-Abelian phase featuring anyonic Majorana excitations.

There exists another scenario in which the intermediate phase is simply a partially polarized phase and smoothly connected to the fully polarized phase. Here, the transition through $B_c$ involves conventional multiparticle excitations due to anisotropic interactions[20]. To resolve these opposing viewpoints, one needs to

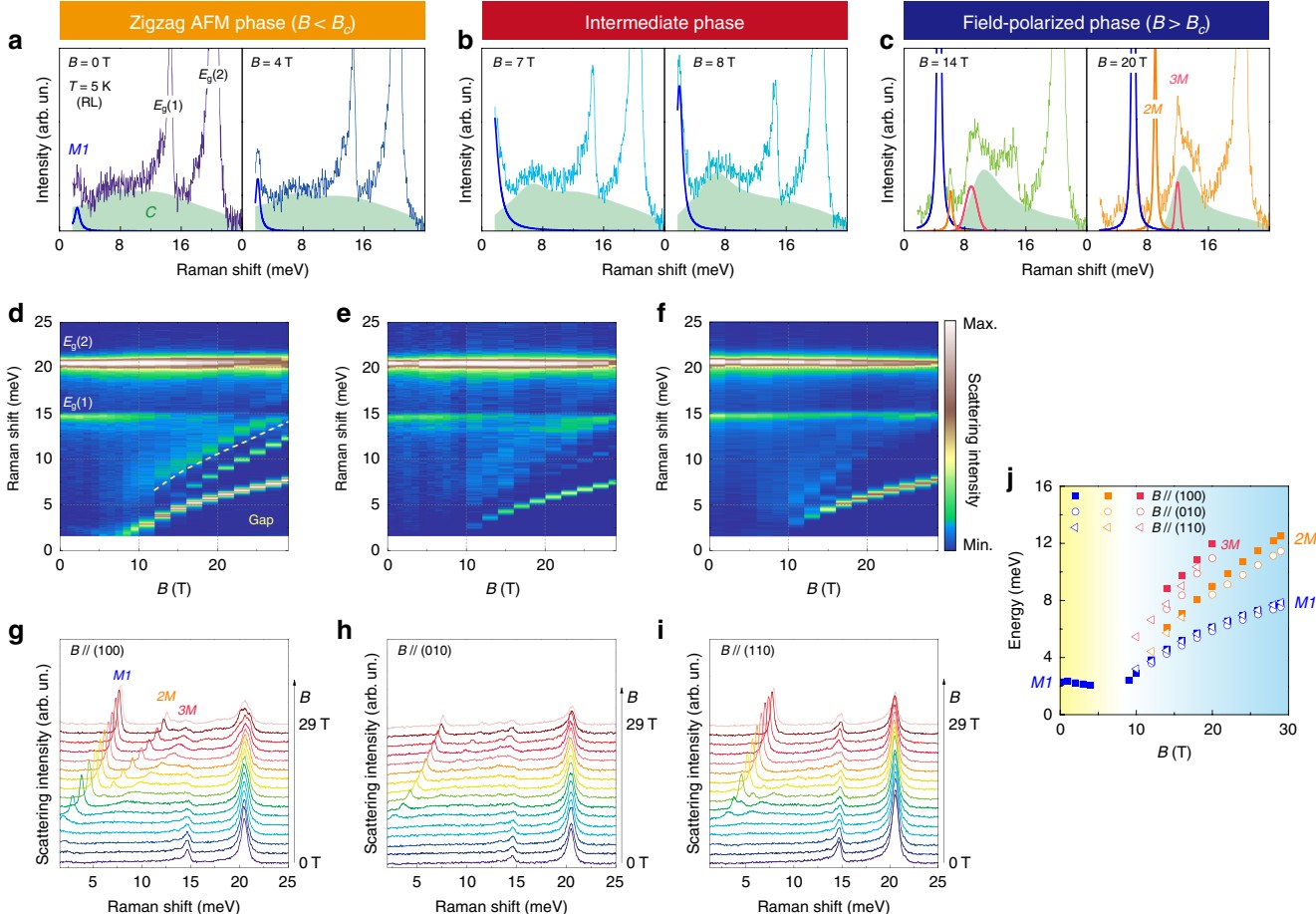

**Fig. 1 Field evolution of magnetic excitations in $\alpha$-RuCl$_3$ through field-induced phases. a–c** As-measured Raman spectra at $T \approx 5$ K. For $H//a$, $\alpha$-RuCl$_3$ passes successively from a zigzag antiferromagnetic- through an intermediate- to a field-polarized-phase with increasing magnetic fields. The color shading denotes the broad continuum ($C$) on top of well-defined sharp peaks of magnetic origin ($M1$, $2M$, $3M$) and phonon modes ($E_g(1)$ and $E_g(2)$). **d–f** Color contour plots of the Raman scattering intensity evolution with magnetic fields aligned along (100), (010), and (110), respectively. The dashed line in d denotes the gap of the fractionalized continuum as a function of field. **g–i** Respective sets of raw Raman data. **j** Field-dependence of the sharp low-energy magnetic excitations compared for different field directions.

clarify the characteristics of quasiparticle excitations emergent in the intermediate-to-high-field phase. So far, inelastic neutron scattering (INS) [21], THz spectroscopy[22], and electron spin resonance (ESR) [23] have revealed a significant reconfiguration of the magnetic response through $B_c$, giving tantalizing evidence for a quantum spin liquid state for magnetic fields just below the fully spin polarized state. These experimental techniques dominantly probe $\Delta S = \pm 1$ excitations. Therefore, complementary experiments sensitive to also singlet ($\Delta S = 0$) excitations are essential for unraveling new aspects of low-energy properties and for obtaining a complete picture of individual quasiparticles.

Here, we employ Raman spectroscopy capable of sensing single- and multiparticle excitations over the sufficiently wide ranges of temperatures $T = 2$–$300$ K, fields $B = 0$–$29$ T, and energies $\hbar\omega = 1$–$25$ meV ($8$–$200$ cm$^{-1}$). At low fields ($B < B_c$) and low temperatures ($T < T_N$) we observe a number of spin-wave excitations superimposed onto a continuum of fractionalized excitations. Towards higher fields above $10$ T, the magnetic continuum opens progressively a gap and its spectral weight is transferred to well-defined sharp excitations that correspond to one-magnon and multimagnon bound states, marking the crossover to a field-polarized phase. In the intermediate phase, a weakly bound state emerges. This bound state is formed via a spectral transfer from the fractionalized continuum through an isosbestic point around $8.75$ meV, and does not smoothly connect to the magnon bound states in the high-field phase. Our results suggest that this weakly bound state carries Majorana characteristics[11,24] and that the intermediate-field phase of $\alpha$-RuCl$_3$ hosts a distinct quantum phase.

## Results

**Field dependence of magnetic excitations.** We performed Raman scattering experiments on oriented single crystals to elucidate the field-evolution of the magnetic excitation spectrum of $\alpha$-RuCl$_3$. A detailed outline of the scattering geometries is given in Supplementary Note 1 and Supplementary Fig. 1, and the full dataset is presented in Supplementary Note 2 and Supplementary Fig. 2. All measurements were carried out with $RL$ circularly polarized light, probing the $E_g$ symmetry channel. Fig. 1a–c shows representative raw spectra obtained at increasing magnetic fields aligned along the crystallographic $a$ axis [$B//(100)$]. We observe two sharp, intense phonon modes at $14.5$ and $20.5$ meV [marked $E_g(1)$ and $E_g(2)$] with a pronounced field dependence, signaling a strong coupling between lattice and spin degrees of freedom (see Supplementary Note 2 and Supplementary Fig. 3 for details). In addition, there are several magnetic excitations with distinct field dependences: at zero field, the magnetic Raman response consists of a broad continuum ($C$; green shading) and a sharp peak ($M1$, blue line). The latter $M1$ peak at $2.5$ meV is assigned to one-magnon scattering arising from a spin flip process by strong spin–orbit coupling and enables us to detect a gap of low-lying excitations at the $\Gamma$-point as a function of field. The $M1$ peak energy and its field dependence matches well recent THz magneto-optical data, confirming the $\Delta S = \pm 1$ scattering process as its origin. The green-shaded continuum $C$ agrees with observations in several previous Raman scattering studies[12,25,26] and has been identified as Majorana fermionic excitations stemming from a fractionalization of spin degrees of freedom in the Kitaev honeycomb model[17]. Although we cannot exclude an incoherent multimagnon contribution to the continuum, the thermal evolution of the continuum follows two-fermionic statistics. Such exotic behavior is not expected for bosonic spin-wave excitations, but rather supports the notion of Majorana fermions[27]. An analysis of the temperature dependence shows that the two-fermionic character increases by about 120% at $B_c = 6.7$ T (when zigzag order is suppressed) compared to $B = 0$ T

(see Supplementary Note 3, and Supplementary Figs. 4 and 5). This can be taken as further evidence for the presence of Majorana fermionic excitations in $\alpha$-RuCl$_3$. As the magnetic field increases above $10$ T, $C$ becomes gapped and its spectral width narrows down. This opening of the gap is traced by the dashed curve in Fig. 1d.

Noteworthy is that the gapped continuum has a finite intensity even at high fields $B > B_c$, while several sharp and well-defined excitations emerge additionally with increasing fields. The residual spectral weight of $C$ at sufficiently high fields means that the excitation spectrum in this regime is not solely exhausted by single- and multiparticle magnons. Indeed, recent numerical calculations of the Kitaev model under applied fields uncovered a wide Kitaev paramagnetic region, reaching far beyond the critical field at finite temperature[28]. We therefore ascribe the gapped continuum excitations to fractional quasiparticles pertinent to the Kitaev paramagnetic state. The $M1$ peak is ubiquitous in all measured fields. The excitation $2M$ (orange line) is split off from the $M1$ peak above $12$ T, while the higher-energy $3M$ feature (red line) appears at the lower boundary of the gapped continuum above $10$–$14$ T. In previous experimental field-dependent studies on $\alpha$-RuCl$_3$ ranging from INS[21], to THz absorption[22], to ESR[23] similar sharp magnetic excitations were reported and interpreted in terms of one-magnon or magnon bound states. In consideration of the narrow spectral form and energy of the corresponding excitations observed in our data, we assign the $2M$ peak to a two-magnon bound state. The $3M$ peak could be either the excitation of a three-magnon bound state, or a van Hove singularity of the gapped continuum. Considering the $3M$ peak sharpens in the high-field regime, the assignment to a three-magnon bound state is more convincing. As the field is lowered to the intermediate phase, the $3M$ excitation is no longer well-defined. The evolution of all magnetic excitations as a function of fields up to $29$ T is depicted in the color contour plots of Fig. 1d–f together with the as-measured Raman spectra in Fig. 1g–i for field directions along (100), (010), and (110), respectively. A slight anisotropy in magnetic excitations as a function of field-direction becomes apparent, which is highlighted in Fig. 1j. In particular, the energy and field ranges of $3M$ are sensitive to the field directions, indicating the presence of non-negligible in-plane anisotropy terms. Our high-field Raman data evidence the coexistence of multimagnon excitations and a gapped continuum that characterizes the quasiparticle landscape of the partially polarized phase. The base temperature is, however, restricted to $T \geq 5$ K in this high-field data. As we show below, a richer spectrum emerges at intermediate fields around $B_c$ upon further cooling.

**Magnetic excitations at intermediate fields.** The reported half-integer thermal Hall conductance[13] is expected from chiral Majorana states along the edges of the 2D honeycomb layers of $\alpha$-RuCl$_3$. Simultaneously, Majorana bound states with possibly anyonic character emerge in the bulk. A detection of these excitations will provide a clue to the topological nature of the field-induced intermediate phase. In the extended Kitaev system, these bound states can occur in different channels[18], namely, from either binding itinerant Majorana fermions to localized fluxes (sketched in Fig. 2a), or by binding two itinerant Majorana fermions (sketched in Fig. 2b). With this in mind, we study the intermediate phase in detail by switching to a magneto-optical cryostat setup, enabling us to reach a base temperature of $T = 2$ K in a field range of $B = 0$–$10$ T. In this setup the sample is tilted by an angle of $18°$ away from the in-plane field geometry, resulting in an additional small but finite out-of-plane field component. Note that here both magnetic field direction as well as light scattering geometry are slightly different from the high-field setup, which

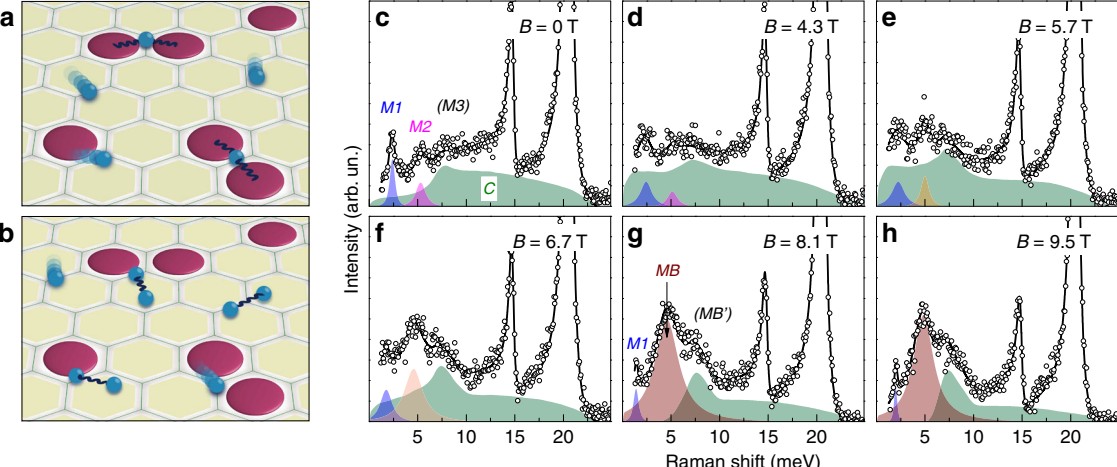

**Fig. 2 Magnetic excitations at intermediate magnetic fields. a** The creation of a bound state from itinerant fermions bound to localized fluxes and **b** from binding only itinerant Majorana fermions. **c–h** Evolution of Raman data obtained at $T = 2$ K (open circles) with increasing fields from 0 to 9.5 T. The shaded regions denote the decomposition of the magnetic excitations into well-defined peaks and a continuum of excitations. The solid line is a sum of all excitations. The *M1* (blue) and *M2* (purple shading) modes at low fields of 0–4.3 T correspond to spin-wave excitations. The mode (*M3*) may be a higher-energy branch of magnon excitations. The excitation *MB* (dark red) above the critical field of 6.7 T is assigned to a Majorana bound state. The shoulder (*MB'*) is either another Majorana bound state or a van-Hove singularity of the Majorana continuum excitations.

prohibits a strict one-to-one comparison of the data. Nonetheless, a good correspondence is found between the high-field *B*//(110) data at $T \approx 5$ K and the magneto-optical data at $T = 9$ K (see Supplementary Note 1). In Fig. 2c–h we inspect the field-dependence of Raman spectra measured at $T = 2$ K. Compared to the $T = 5$ K high-field data shown in Fig. 1a, the $T = 2$ K data show new sharp *M2* and (*M3*) structures at 5 and 7.5 meV, respectively, in addition to the one-magnon excitation *M1* and the fractionalized continuum *C*. The *M2* peak may be tentatively assigned to a two-magnon-like excitation in the singlet sector. Unlike conventional two-magnon scattering, the *2M* peak is narrow possibly because a large portion of the two-magnon excitation decays into the continuum. As the field increases, the respective modes evolve in a disparate manner. Initially (at $B = 0$–4.3 T), the *M1* and *M2* modes are slightly suppressed, while the continuum *C* is partially renormalized toward lower energies (Fig. 2c, d). As $B_c$ is approached and through 6.7 T, the spectral weight of the continuum is massively redistributed (Fig. 2e, f). A new low-energy *MB* mode evolves from the low-field *M2* mode and a shoulder structure (*MB'*) appears out of the (*M3*) peak. Apparently, the continuum of Majorana excitations is gapped above 8.1 T (Fig. 2g, h). A recent INS study reported a similarly broad, emerging excitation in the intermediate field-induced phase[21]. It was tentatively discussed as a possible Majorana bound state, but an ultimate assignment was hindered by the lack of a detailed temperature study.

**Formation of a Majorana bound state**. To analyze the field-induced spectral weight redistribution carefully, we replot phonon-subtracted Raman data taken at $T = 2$ K in Fig. 3a. With increasing field a distinct transfer of spectral weight from the mid-energy (green-shaded *A2*) to the low-energy regime (purple-shaded *A1*) is observed, with an isosbestic point located around $\omega_{iso} = 8.75$ meV, at which the magnetic Raman response is independent of the external field. The systematic field-induced redistribution of spectral weight through this isosbestic point suggests an intimate connection between the continuum and the newly formed excitation, and therefore supports the formation of a low-energy Majorana bound state (*MB*) through a confinement of the high-energy broad continuum of Majorana fermionic

excitations. We also note that the *MB* mode appears in the same symmetry channel as the continuum of Majorana fermionic excitations (see Supplementary Note 4 and Supplementary Fig. 6). Consistent with the high-field data presented in Fig. 1, we observe a remaining intensity of the continuum *C* of Majorana fermions at 9.5 T, i.e., corresponding to the nontrivial quantum phase. A coexistence of massive Majorana fermions that form a broad, gapped continuum together with a sub-gap *MB* state is not compatible with the trivial polarized phase that is characterized by the multimagnon bound states. Figure 3b shows the thermal evolution obtained at $B = 9.5$ T (see Supplementary Note 5, and Supplementary Fig. 7 for full data set). As the temperature increases, the low-energy mode *MB* gradually loses in intensity and shifts toward higher energies. Meanwhile, the continuum *C* slightly gains in intensity. The field- and thermal evolution of magnetic modes is visualized in a contour plot in Fig. 3c, based on fits to the as-measured data. It especially highlights the similar evolution of *MB* with field and temperature: around 5 meV its spectral weight starts to appear at 4.7 T and grows with increasing field, while it becomes thermally stabilized upon cooling below 12 K at 9.5 T. This suggests that quantum and classical fluctuations play the same role in a confinement-deconfinement transition due to a proximate Kitaev paramagnetic state at elevated temperature[11].

Fig. 3d plots the energy of the *MB* mode as a function of field together with the spin-wave excitations observed in the zigzag ordered phase as well as in the high field spin-polarized phase. Here, full squares, empty circles, and empty triangles denote *B*//(100), *B*//(010), and *B*//(110), respectively. We note a smooth transition from *M2* to *MB* through $B_c$. This weak field-dependence suggests that the *MB* mode corresponds to an excitation in the singlet sector ($\Delta S_z = 0$), to which Raman spectroscopy is a natural probe. As the magnon corresponds to a condensation of Majorana fermions, the *M2*-to-*MB* mode evolution may be interpreted in terms of a condensation-to-confinement crossover where the magnon excitations observed at low field gradually evolve into Majorana bound states in the singlet sector. Since the continuum *C* of deconfined Majorana excitations above $B_c$ is massively gapped with an onset energy of 4–6 meV (see Figs. 1d–f and 2g, h), the low-energy bound state

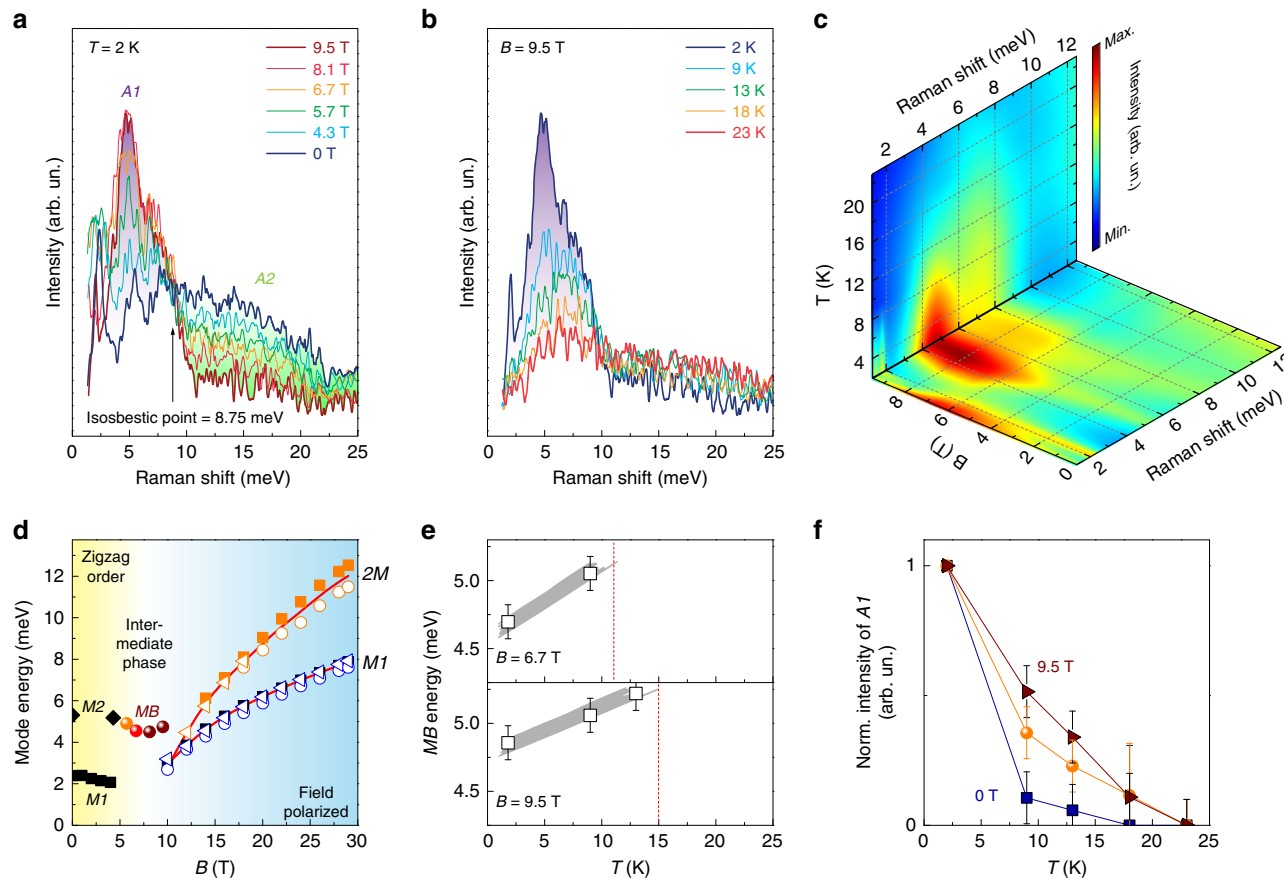

**Fig. 3 Spectral weight redistribution and formation of a bound state through $B_c$. a** Raman spectra obtained at $T = 2$ K with increasing magnetic fields. **b** Raman spectra at $B = 9.5$ T recorded with increasing temperatures. Phonon modes are subtracted in (**a**) and (**b**) for better visualization of the magnetic excitations. **c** Contour plot of the magnetic Raman scattering intensity as a function of temperature and field. **d** Field-dependence of excitations around the intermediate phase (spheres) together with spin-wave excitations at low and high magnetic fields. **e** Thermal evolution of the $MB$ mode energy at $B = 6.7$ and 9.5 T. Gray lines are guides to the eyes. **f** Thermal melting of the low-temperature magnetic modes at 0, 6.7, and 9.5 T. Standard deviations in (**e**, **f**) are indicated by error bars.

can be created within the gap due to confinement. Unlike the $M2$-to-$MB$ mode crossover below $B_c$, the $MB$ mode is not smoothly linked to the $2M$ bound state for fields above $B_c$. Rather, as the field increases, the $2M$ mode splits from the $M1$ mode, and both excitations are observed prominently. This suggests that the high-field phase stabilizes conventional quasiparticles against the fractional excitations. Interestingly, a signature of the $MB$ mode remains absent in our data obtained from the high-field setup, as well as in data from a related recent high-field Raman and THz study[29], implying that the parameters temperature, scattering geometry, and magnetic field direction are decisive in stabilizing Majorana bound states. It also suggests that we cannot assert a direct relation between the two excitations $M2$ (at low fields) and $2M$ (at high fields). Our data is also contrasted by the rather smooth transition of quasiparticle excitations observed in ESR experiments through 10 T[23], due to the different selection rules for quasiparticle excitations in Raman vs. ESR. The discontinuous evolution observed through 10 T in our data admittedly may be expected due to the slightly different experimental conditions between the high-field and the magneto-cryostat setups. Yet, the small temperature difference of $\approx 3$ K can hardly account for a jump of 1 meV. We also recall that the vanishing thermal Hall conductance around 10 T parallels the disappearance of the MB mode[13].

Further support of the $MB$ state interpretation comes from the temperature dependence at two different magnetic fields, 6.7 and

9.5 T (see Fig. 3e and Supplementary Figs. 8 and 9). We see a clear initial increase in the $MB$ mode energy at both fields as the temperature rises. This is contrasted by conventional (magnon) unbound excitations, which continuously soften with increasing temperature. For bound states, however, the thermal energy competes with the binding energy[30], until eventually a thermally induced unbinding takes place. We can trace the energy of the $MB$ excitation up to temperatures of 9 K (for $B = 6.7$ T) and 13 K (for $B = 9.5$ T). At higher temperatures the $MB$ quasiparticles melt and decay into the higher-energy continuum, and therefore become ill-defined. Based on the peak-energy shift observed in the narrow, limited temperature window, we estimate a lower limit of the binding energy of 0.5 meV. Estimating the binding energy from the gap size yields a slightly higher value of about 1 meV (see Supplementary Note 6). The thermal evolution of area $A1$, summarized in Fig. 3f, highlights the gradual melting of the bound state at 9.5 T (dark red triangles) with increasing temperature, while the conventional magnetic excitations at 0 T (blue squares) abruptly vanish above $T_N$. All these observations are consistent with the picture of a quasi-bound state of Majorana fermions that is pulled below the gapped fractionalized continuum by residual interactions of non-Kitaev origin[31].

## Discussion

In the Kitaev model, Majorana bound states are created through flux pairs combined with Majorana fermions in a $\Delta S_z = \pm 1$

channel[17]. However, as flux excitations are largely invisible to the Raman scattering process[32], the bound states between the flux and Majorana fermions barely contribute to the magnetic Raman signal. In the presence of additional non-Kitaev terms, the creation of bound states from itinerant Majorana fermions is enhanced (see the cartoon in Fig. 2b)[18]. As Raman scattering probes mainly the $\Delta S_z = 0$ channel, we conclude that the $MB$ mode largely consists of the latter Majorana singlet bound states. This interpretation is supported by the smooth crossover from the $M2$ magnon mode to the bound state $MB$ through $B_c$ (see Fig. 3d), as both excitations arise from a spin-conserving scattering process. In such a case, $\alpha$-RuCl$_3$ as an apt realization of the $K - \Gamma$ model (see Supplementary Note 6) can host an exotic intermediate-field phase. In relation to this issue, we mention that a numerical study of the $J - K - \Gamma - \Gamma'$ model shows an extended regime of a chiral spin liquid for the out-of-plane field. Once the magnetic field is tilted significantly toward the in-plane direction, the intermediate topological phase vanishes[33]. This discrepancy raises the challenging question whether the recently reported field-induced phase has a non-Abelian nature and how the in-plane intermediate phase transits to the alleged chiral spin-liquid phase, if the intermediate phase is of topologically trivial nature. As the Chern number changes its sign as a function of the magnetic field direction, the field-angular variation of magnetic excitations should be detailed.

Our finding demonstrates that a non-trivial crossover from the zigzag through the intermediate to the high-field phase involves a strong reconfiguration of the fractionalized continuum excitations, calling for future work to shed light on the relation between the observed Majorana bound states in an in-plane intermediate field phase of $\alpha$-RuCl$_3$ and the non-Abelian phase predicted for out-of-field directions.

## Methods

**Crystal growth**. Single crystals of $\alpha$-RuCl$_3$ were synthesized by a vacuum sublimation method. A commercial compound of RuCl$_3$ (Alfa Aesar) was ground and dried in a quartz tube under vacuum in order to fully dehydrate. The evacuated quartz tube was sealed and placed in a temperature gradient furnace. A powder of RuCl$_3$ was heated at 1080 °C for 24 h and then slowly cooled down to 600 °C at a rate of $-2$ °C h$^{-1}$. The resulting single crystals have typical sizes of about 5 mm × 5 mm × 0.5 mm, with a shiny black surface. Their thermodynamic and spectroscopic properties have been thoroughly characterized[9,12,15,16].

**Raman scattering**. High magnetic fields up to 29 T were generated using the resistive magnet M10 at the LNCMI Grenoble. The sample was kept at a temperature $T \approx 5$–10 K and illuminated with a 515 nm solid state laser (ALS Azur Light Systems) at a laser power $P = 0.05$ mW and a spot size of 3 μm diameter. Resulting Raman spectra were collected in Voigt geometry for in-plane fields, and in Faraday geometry for out-of-plane geometry, using volume Bragg filters (OptiGrate) in transmission geometry and a 70 cm focal distance Princeton Instruments spectrometer equipped with a liquid N$_2$ cooled Pylon CCD camera.

Temperature-dependent Raman scattering experiments in intermediate fields of $B = 0$–9.5 T were carried out in 90° scattering geometry using a Horiba T64000 triple spectrometer equipped with a Dilor Spectrum One CCD and a Nd:YAG laser emitting at $\lambda = 532$ nm (Torus, Laser Quantum). A $\lambda/4$-plate was used to generate left- and right-circularly polarized light ($RL$). The laser power was kept to $P = 4$ mW with a spot diameter of about 100 μm to minimize heating effects. A base temperature of $T_{base} = 2$ K was achieved by fully immersing the sample in superfluid He. Measurements at elevated temperatures were carried out in He gas atmosphere. From a comparison between Stokes- and anti-Stokes scattering we estimate the laser heating to be of 3 K within the He gas environment. The sample temperatures are corrected accordingly. In-plane magnetic fields were applied via an Oxford Spectromag split coil system ($T = 2$–300 K, $B_{max} = 10$ T).

**Data analysis**. The mid-energy regime of the continuum of Majorana fermionic excitations observed in Raman spectroscopy arises from the simultaneous creation or annihilation of a pair of Majorana fermions. Its temperature dependence can be described by two-fermionic statistics, $I_{MF} = [1 - f(\epsilon_1)][1 - f(\epsilon_2)]\delta(\omega - \epsilon_1 - \epsilon_2)$; with $f(\epsilon) = 1/[1 + e^{\epsilon/k_B T}]$ (see ref. [27] for details). Additional terms that stem from deviations of the pure Kitaev model ($\Gamma$-term, Heisenberg exchange coupling) culminate in an additional bosonic background term, $I_B = 1/[e^{\epsilon/k_B T} - 1]$. The thermal

evolution of the continuum has been fitted to a superposition of both contributions.

Fits to the phonon spectrum were applied using symmetric Lorentzian lineshapes, as well as asymmetric Fano lineshapes[34] [$I(\omega) = I_0 \frac{(q+\epsilon)^2}{(1+\epsilon^2)}$, with $\epsilon = (\omega - \omega_0)/\Gamma$, and $\Gamma =$ full width at half maximum] in case of a strong coupling between lattice- and spin degrees of freedom. The parameter $1/|q|$ characterizes the degree of asymmetry and—consequently—gives a measure of the coupling strength.

## Data availability

The authors declare that all data supporting the findings of this study are available as plots within the article and its Supplementary Information file. Raw datasets generated during and/or analyzed during the current study are available from the corresponding authors on reasonable request.

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

## Acknowledgements

We acknowledge important discussions with Natalia Perkins, Yuji Matsuda, and Stephen Nagler. Part of this work was performed at the LNCMI, a member of the European Magnetic Field Laboratory (EMFL). This work was supported by "Niedersächsisches Vorab" through the "Quantum- and Nano-Metrology (QUANOMET)" initiative within the project NL-4, DFG-Le967-16, and the Excellence Cluster DFG-EXC 2123 Quantum Frontiers. The work at CAU was supported by the National Research Foundation (NRF) of Korea (Grant no. 2020R1A2C3012367).

## Author contributions

K.-Y.C. together with Y.G. and P.L. conceived and designed the experiments. S.-H.D., Y.C., C.H.L. and K.-Y.C. synthesized the single crystals. D.W., Y.C., C.F. and Y.G. performed the Raman spectroscopic experiments. Y.C., K.-Y.C., D.W., Y.G., C.F. and P.L. analyzed the data. D.W., K.-Y.C., Y.G., P.L. and Y.C. participated in the writing of the paper. All authors discussed the results and commented on the paper.

## Competing interests

The authors declare no competing interests.
