## [Peer Review File · Nature Communications]

Reviewers' comments:

Reviewer #1 (Remarks to the Author):

In their manuscript “magnon bound states vs. anyonic Majorana excitations in the Kitaev honeycomb magnets α -RuCl₃” by Wulferding et al. provides systematic Raman spectroscopic study in the Kitaev honeycomb magnets α -RuCl₃ under magnetic field by using both high field facility and lab-based cryostat system. At low fields and low temperature, two magnon peaks were observed along with the Raman continuum originated from the fractional spin excitations; at high field three sharp peaks which were assigned as one-magnon and magnon bound states were detected. In the intermediate phase, a broad peak at around 5 meV was formed through a spectral transfer from the fractionalized continuum, which displays Majorana characteristics and was assigned as the Majorana bound states. Recent experimental results showed that the in-plane magnetic field can suppress the AFM order and produce a topological quantum spin liquid state before the fully spin polarized state in α -RuCl₃. This work provides another evidence for this scenario.

Questions and comments

1. In figure 2c, the Raman continuum (the green shade) has a peak component centered at around 7.5 meV, which has similar width with the Majorana bound state peak at around 5 meV. This peak becomes clearer at the intermediate phase (from 6.7 to 9.5 T). Can this peak feature to be assigned as another Majorana bound state? Or it comes from the van Hove singularity.
2. In Figure 1, the M2' can only be resolved in spin polarized phase, but the M2 can be observed in AFM phase in figure2, which is probably due to difference of measurement configuration. Another possibility is that they have different origins. Please give some reasons for assigning them to the same M2 peak.
3. In figure 1e, the phonon peak at around 20 meV which is associated with the in-plane movement of Ru atoms shows some kind of splitting at high field. Please clarify this phenomena is genuine or some kind of measurement error. If it is true, there might be some kind of structure distortion induced by magnetic field. Please give some discussion about it.
4. In figure 3e, the energy jumping for the Majorana bound state provides evidence of unbinding, but the energy changes are harder to read from the spectra. Could you provide detailed fitting for exacting the energy of the Majorana bound state in the supplementary?

I would like to recommend this manuscript to be published in Nature communications after the questions are clearly addressed.

Reviewer #2 (Remarks to the Author):

This manuscript presents in-field Raman scattering measurements on the Kitaev spin liquid candidate α -RuCl₃. The authors are particularly interested in the intermediate-field phase between 7 T and 10 T whose precise nature is under debate. By studying the field evolution of the spin response probed by Raman scattering, they identify a broad (but well defined) peak in the intermediate-field spectrum as a bound state of anyonic spin-liquid excitations, suggesting that the intermediate-field phase is a quantum spin liquid.

I think the measurements presented in the manuscript could be interesting for a large community of scientists working on Kitaev magnets. However, while I largely agree with the interpretation of the results, I feel that it has room for improvement. Therefore, I suggest the publication of the manuscript after the authors address the following questions / comments:

1. In the third paragraph of Sec. II, the authors claim that they are looking for “anyonic Majorana bound states”, like those shown in Figs. 2(a,b). Later, they identify the MB mode in their Raman response as such a bound state. However, those bound states consisting of either two Majorana fermions or a flux pair and a Majorana fermion are not actually anyonic; while they are composed of anyons, they themselves are topologically trivial (i.e., not anyons). Local probes like Raman scattering cannot directly couple to anyons and, therefore, a sharp mode in a Raman spectrum can never correspond to an anyon.

2. The authors seem to identify the M2 mode in the low-field Raman response as a multi-magnon contribution. Given that this mode is rather sharp, would it not be more natural to identify it as a primarily single-magnon contribution (with perhaps some multi-magnon component)? Since the low-field zigzag order enlarges the magnetic unit cell, there should be distinct magnon branches that could plausibly correspond to the M1 and M2 modes. Also, from the spin-fractionalization rule in the Kitaev honeycomb model, $\sigma_j^z = i b_j^z c_z$, the bound state in Fig. 2(a) is actually equivalent to a magnon (in terms of both symmetry and topology) and, therefore, the crossover between the M2 and MB modes could be naturally understood.

3. To decide if the continuum feature in their Raman response is a Majorana continuum or a multi-magnon continuum, the authors consider the thermal evolution of the response and fit it with a superposition of a Bose function (corresponding to magnons) and a product of two Fermi functions (corresponding to Majorana pairs). I think it is very encouraging that the second contribution is stronger at intermediate fields than at low fields. However, I feel that this result could be even stronger if the authors were a bit more quantitative and provided actual numbers for the coefficients of the two contributions at both 0 T and 6.7 T. I am also wondering if it is possible to do the same fitting at high fields where the first contribution, corresponding to magnons, would presumably become more dominant.

4. In Fig. 3(e), the authors plot the energy of the MB mode as a function of temperature and attribute a sudden drop in the MB energy to an unbinding of a bound state. However, should not such an unbinding correspond to an increase in the energy? Also, is the sudden drop of the MB energy accompanied by a sudden broadening of the MB peak? If a topologically trivial bound state unbinds into its anyonic constituents, the corresponding sharp mode should disperse into a continuum.

Finally, if I understand it correctly, there should be no connection between the M3' mode in Fig. 1(a) and the M3' mode in Fig. 2(c). Would it not be better to use different labels for them?

REFEREE 1

In their manuscript “magnon bound states vs. anyonic Majorana excitations in the Kitaev honeycomb magnets α -RuCl₃” by Wulferding et al. provides systematic Raman spectroscopic study in the Kitaev honeycomb magnets α -RuCl₃ under magnetic field by using both high field facility and lab-based cryostat system. At low fields and low temperature, two magnon peaks were observed along with the Raman continuum originated from the fractional spin excitations; at high field three sharp peaks which were assigned as one-magnon and magnon bound states were detected. In the intermediate phase, a broad peak at around 5 meV was formed through a spectral transfer from the fractionalized continuum, which displays Majorana characteristics and was assigned as the Majorana bound states. Recent experimental results showed that the in-plane magnetic field can suppress the AFM order and produce a topological quantum spin liquid state before the fully spin polarized state in α -RuCl₃. This work provides another evidence for this scenario.

Questions and comments

1. In figure 2c, the Raman continuum (the green shade) has a peak component centered at around 7.5 meV, which has similar width with the Majorana bound state peak at around 5 meV. This peak becomes clearer at the intermediate phase (from 6.7 to 9.5 T). Can this peak feature to be assigned as another Majorana bound state? Or it comes from the van Hove singularity.

AUTHORS: We thank the Referee for raising this issue. As the Referee correctly pointed out, there is a 7.5 meV kink in the data at zero field, which becomes prominent in the intermediate phase. As this (MB') feature appears in the shoulder of the MB state, it can be interpreted in terms of another Majorana bound (MB') state, which evolves from the higher-energy branch of the magnon excitation (M3).

However, we cannot rule out the possibility that this 7.5 meV feature is merely due to a van-Hove singularity of the continuum just at the edge of the gap. We realize that our discussion of the (M3) feature in the manuscript is a bit vague and we improved it accordingly. The parentheses in both (MB') and (M3) are used to stress that the assignment is tentative for the 7.5 meV case.

2. In Figure 1, the M2' can only be resolved in spin polarized phase, but the M2 can be observed in AFM phase in figure2, which is probably due to difference of measurement configuration. Another possibility is that they have different origins. Please give some reasons for assigning them to the same M2 peak.

AUTHORS: Here we would first like to point to the fact that the data presented in Fig. 2 has been obtained at $T = 2$ K, while the data in Fig. 1 has been obtained at a higher temperature, just below T_N (since the distinct M1 mode is still weakly observable). We suggest that this temperature difference is the reason for the different observations presented in Fig 1 and Fig 2.

Our assignment of M2 at $B < B_c$ is based on previous experiments using inelastic neutron scattering (Banerjee, et al., npj Quantum Mater. 3, 8), THz absorption (Wang, et al., PRL 119, 227202), and ESR (Ponomaryov, et al., PRB 96, 241107), where a magnetic excitation at $E \sim 2.6$ meV for $B = 0$ is reported, thus matching well a two-magnon Raman scattering signal at ~ 5.3 meV as observed in our data. At higher fields a sharp mode (re-)emerges, labelled as 2M. As summarized in Fig. 3d, the M2 mode is connected to the MB mode in the intermediate phase, but not to the M2' mode in the high-field regime because it is a two-magnon bound state. Therefore it is labelled 2M. This relabelling will avoid any confusion to the readers.

3. In figure 1e, the phonon peak at around 20 meV which is associated with the in-plane movement of Ru atoms shows some kind of splitting at high field. Please clarify this phenomena is genuine or some kind of measurement error. If it is true, there might be some kind of structure distortion induced by magnetic field. Please give some discussion about it.

AUTHORS: We thank the Referee for pointing us towards this observation. As the main focus of our manuscript is on magnetic excitations, we didn't emphasize too much on the phonon behavior. After a careful analysis we noticed that even at lowest fields there is a faint high-energy shoulder at the 20 meV phonon (see blue fitting curve in Supplementary Fig. S3). As this E_g phonon is twofold degenerate, a splitting might indicate a weak structural distortion (related to stacking faults) lifting the degeneracy. With increasing fields this shoulder becomes more pronounced, although it does not show any distinct energy shift, signaling that the structure is not affected by increasing magnetic fields. Distinct from the higher-energy shoulder, a second contribution is seen in the lower-energy side, which approaches the 20 meV

phonon at elevated fields and eventually merges with the phonon. Due to its different behavior regarding magnetic fields, mode intensity, and linewidth, we assign it to a magnetic excitation within the continuum C. Noticeably, this excitation locks in energetically with the phonon around $B = 22$ T. Around this field the phonon intensity decreases and the linewidth increases (see also color contour plot in Fig. S2), pointing towards an enhanced interaction between lattice- and spin degrees of freedom (i.e., a magnon-like excitation that evolves out of the gapped continuum excitations for magnetic fields close to saturation). As the E_g phonons show Fano resonances, the field-induced intensity variation is largely due to the changing strength of the phonon-continuum coupling. We added a paragraph and Fig. S3 to the Supplementary to better describe these observations.

4. In figure 3e, the energy jumping for the Majorana bound state provides evidence of unbinding, but the energy changes are harder to read from the spectra. Could you provide detailed fitting for exacting the energy of the Majorana bound state in the supplementary?

AUTHORS: We added a detailed plot including the individual fitting curves to the Raman data in the supplement accordingly (see supplementary section S6). The subtle energy change as a function of temperature is related to a thermally induced unbinding of the MB state into a continuum. Since the MB quasiparticles are no longer well-defined above the melting temperature, we will not try to trace the peak energy, for example, above 15 K at $B=9.5$ T.

I would like to recommend this manuscript to be published in Nature communications after the questions are clearly addressed.

AUTHORS: We thank the Referee for his/her positive comments and valuable suggestions.

REFEREE 2

This manuscript presents in-field Raman scattering measurements on the Kitaev spin liquid candidate α -RuCl₃. The authors are particularly interested in the intermediate-field phase between 7 T and 10 T whose precise nature is under debate. By studying the field evolution of the spin response probed by Raman scattering, they identify a broad (but well defined) peak in the intermediate-field spectrum as a bound state of anyonic spin-liquid excitations, suggesting that the intermediate-field phase is a quantum spin liquid.

I think the measurements presented in the manuscript could be interesting for a large community of scientists working on Kitaev magnets. However, while I largely agree with the interpretation of the results, I feel that it has room for improvement. Therefore, I suggest the publication of the manuscript after the authors address the following questions / comments:

AUTHORS: We thank the Referee for his/her positive comments and the valuable suggestions.

1. In the third paragraph of Sec. II, the authors claim that they are looking for “anyonic Majorana bound states”, like those shown in Figs. 2(a,b). Later, they identify the MB mode in their Raman response as such a bound state. However, those bound states consisting of either two Majorana fermions or a flux pair and a Majorana fermion are not actually anyonic; while they are composed of anyons, they themselves are topologically trivial (i.e., not anyons). Local probes like Raman scattering cannot directly couple to anyons and, therefore, a sharp mode in a Raman spectrum can never correspond to an anyon.

AUTHORS: We thank the Referee for raising this issue. We would like to point to the key finding of our work, which is the identification of a MB state in the narrow intermediate field differentiated from the partially polarized phase by the quasiparticle characteristics. As the Referee pointed out, the observed MB state should not be regarded as anyons predicted in the original Kitaev model. Although the Raman scattering data alone are not sufficient to verify the existence of Majorana anyons, our work demonstrates the exotic nature of the excitation spectrum in the intermediate field range that contains confined Majorana fermions. We have improved the wording and clearly state this in the revised version of our manuscript. We believe that our experimental observation of the MB related excitations greatly adds to the understanding of the formation of Majorana anyons in candidate materials and lays the foundation to stabilize Majorana anyons. A possible route towards experimental realization and controlled braiding is to adopt graphene/ α -RuCl₃ and metal/ α -RuCl₃ heterostructures to control anyons through electrical currents and interface strain.

2. The authors seem to identify the M2 mode in the low-field Raman response as a multi-magnon contribution. Given that this mode is rather sharp, would it not be more natural to identify it as a primarily single-magnon contribution (with perhaps some multi-magnon component)? Since the low-field zigzag order enlarges the magnetic unit cell, there should be distinct magnon branches that could plausibly correspond to the M1 and M2 modes. Also, from the spin-fractionalization rule in the Kitaev honeycomb model, $\sigma_j^z = i b_j^z c_z$, the bound state in Fig. 2(a) is actually equivalent to a magnon (in terms of both symmetry and topology) and, therefore, the crossover between the M2 and MB modes could be naturally understood.

AUTHORS: We agree with the Referee that solely based on our data it is difficult to clearly assign the M2 mode around 5 meV to a two-magnon (singlet) scattering process or to a one-magnon (triplet) mode. It is also correct that the narrow linewidth of M2 is in line with a single-magnon contribution and that a one-magnon mode would naturally connect to a Majorana bound state composed of a localized flux and itinerant fermions. However, we are inclined to assign M2 to a two-magnon-like component: The magnetic excitation spectrum at these intermediate energies is not well-described by either Majorana or a spin-wave picture. In this energy range, single spin-waves would be destabilized to the multi-spin-wave or Majorana continuum. The sharp M2 peak may be part of a two-magnon excitation that does not decay into the continuum. This preference is based on the selection rule ($\Delta S_z=0$) of Raman scattering. The MB mode appears in the singlet sector (corresponding to bound itinerant Majorana fermions), which would naturally connect to a singlet (two-magnon-type) excitation in the low-field phase.

3. To decide if the continuum feature in their Raman response is a Majorana continuum or a multi-magnon continuum, the authors consider the thermal evolution of the response and fit it with a superposition of a Bose function (corresponding to magnons) and a product of two Fermi functions (corresponding to Majorana pairs). I think it is very encouraging that the second contribution is stronger at intermediate fields

than at low fields. However, I feel that this result could be even stronger if the authors were a bit more quantitative and provided actual numbers for the coefficients of the two contributions at both 0 T and 6.7 T. I am also wondering if it is possible to do the same fitting at high fields where the first contribution, corresponding to magnons, would presumably become more dominant.

AUTHORS: We thank the Referee for the suggestion to quantify the ratio of bosonic to fermionic excitations, which will help to underline the field effects. Therefore, we included a new Fig. S5 in the supplementary (see also below) and added a comment accordingly to supplementary section S3. We find that the fermionic-to-bosonic ratio is enhanced by about 120% when approaching a critical field. We also agree that it would be very insightful to add a third temperature dependence at high magnetic fields $B > B_c$. Unfortunately, until now, no variable temperature environment is available at the high-field facilities.

4. In Fig. 3(e), the authors plot the energy of the MB mode as a function of temperature and attribute a sudden drop in the MB energy to an unbinding of a bound state. However, should not such an unbinding correspond to an increase in the energy? Also, is the sudden drop of the MB energy accompanied by a sudden broadening of the MB peak? If a topologically trivial bound state unbinds into its anyonic constituents, the corresponding sharp mode should disperse into a continuum.

AUTHORS: We thank the Referee for highlighting this issue. As the Referee rightly pointed out, for example, the MB energy of the $B=9.5$ T spectra increases with increasing temperature up to about 13 K. This is a thermal unbinding process. In Fig. 3e of the original version, the MB energy undergoes a sudden drop on heating through 15 K. However, defining a MB peak energy at these high temperatures is too ambiguous due to a structured continuum (see Figs. S8 and S9): As the MB state undergoes a thermally induced unbinding into a continuum for temperatures above 15 K, the quasiparticle energy is experimentally not clearly traceable over this wide temperature range. In consideration of this, it is fair to present the center energy of the MB state only for temperatures below 15 K (see the revised Figure 3e).

Finally, if I understand it correctly, there should be no connection between the $M3'$ mode in Fig. 1(a) and the $M3'$ mode in Fig. 2(c). Would it not be better to use different labels for them?

AUTHORS: We thank the Referee for pointing us to this improved labeling. We replaced the intermediate-field $M3'$ with (MB') in Fig. 2. As answered in the reply to the first Referee, two interpretations are possible for the (MB') peak: another MB excitation vs a van-Hove singularity. We clearly state this in the revised version. We do assign the $M3'$ feature (now labeled as $3M$ in Fig. 1) to the putative three-magnon bound (now labeled as $3M$) state which lies on the onset of the gapped of the continuum of Majorana fermionic excitations. At sufficiently high fields, the attractive interactions between quasiparticles are strong enough to form the three-magnon bound state. We clarified this issue to avoid any confusion to the readers.

LIST OF CHANGES (Bold = revised):

(1)

Affiliation 4

“LNCMI, CNRS, EMFL, Univ. Grenoble Alpes, 38000 Grenoble, France”

to

“Univ. Grenoble Alpes, INSA Toulouse, Univ. Toulouse Paul Sabatier, EMFL, CNRS, LNCMI, 38000 Grenoble, France”

(2)

Abstract

“Here we present Raman spectroscopic data revealing low-energy quasiparticles...”

to

“Here we present Raman spectroscopic data revealing novel low-energy quasiparticles...”

(3)

Introduction

“To resolve these opposing scenarios, one needs to clarify the nature of quasiparticle excitations emergent in the intermediate-to-high-field phase.”

to

“To resolve these opposing scenarios, one needs to clarify the characteristics of quasiparticle excitations emergent in the intermediate-to-high-field phase.”

(4)

Introduction

“...a significant reconfiguration of the magnetic response through B_c . These methods generally probe $\Delta S = \pm 1$ excitations.”

to

“...a significant reconfiguration of the magnetic response through B_c , giving tantalizing evidence for a quantum spin liquid state for magnetic fields just below the fully spin polarized state. These experimental techniques dominantly probe $\Delta S = \pm 1$ excitations.”

(5)

Introduction

“...the magnetic continuum opens progressively a gap and its spectral weight is transferred to well-defined sharp excitations that correspond to one-magnon and magnon bound states, marking the crossover to a field-polarized phase.”

to

“...the magnetic continuum opens progressively a gap and its spectral weight is transferred to well-defined sharp excitations that correspond to one-magnon and multimagnon bound states, marking the crossover to a field-polarized phase.”

(6)

Fig. 1 Caption

“The color shading denotes the broad continuum (C) on top of well-defined sharp peaks (M1, M2', M3) and phonon modes ($E_g(1)$ and $E_g(2)$).”

to

“The color shading denotes the broad continuum (C) on top of well-defined sharp peaks of magnetic origin (M1, 2M, 3M) and phonon modes ($E_g(1)$ and $E_g(2)$).”

Added sentence:

“The dashed line in b denotes the gap of the fractionalized continuum as a function of field.”

(7)

Results

“We performed Raman scattering experiments on oriented single crystals to elucidate the field-evolution of the magnetic excitation spectrum of α -RuCl₃ (see Supplementary Information, section S1 for a detailed outline of the scattering geometries and section S2 for the full dataset).”

to

“We performed Raman scattering experiments on oriented single crystals to elucidate the field-evolution of the magnetic excitation spectrum of α -RuCl₃. A detailed outline of the scattering geometries is given in section S1, Supplementary Information, and the full dataset is presented in section S2.”

(8)

Results

“Fig. 1a shows representative raw spectra obtained at increasing fields aligned along the crystallographic a axis [B // (100)]. Besides two sharp, intense phonon modes at 14.5 meV and 20.5 meV [marked E_g(1) and E_g(2)], we observe several magnetic excitations with a pronounced field dependence:”

to

“Fig. 1a shows representative raw spectra obtained at increasing magnetic fields aligned along the crystallographic a axis [B // (100)]. We observe two sharp, intense phonon modes at 14.5 meV and 20.5 meV [marked E_g(1) and E_g(2)] with a pronounced field dependence, signaling a strong coupling between lattice and spin degrees of freedom (see Supplementary Information, Fig. S3 for details). In addition, there are several magnetic excitations with distinct field dependences:”

(9)

Results

“The latter M1 excitation at 2.5 meV...”

to

“The latter M1 peak at 2.5 meV...”

(10)

Results

“The M1 mode energy and its field dependence matches well...”

to

“The M1 peak energy and its field dependence matches well...”

(11)

Results

“As detailed in Supplementary Information, section S3, a more prominent two-fermionic character is present at B_c = 6.7 T (when zigzag order is suppressed) compared to B = 0 T. This can be taken as further evidence for the presence of Majorana fermionic excitations in α -RuCl₃. As the magnetic field increases above 10 T, C becomes gapped and its spectral width narrows down. This leads to a build-up of spectral weight towards the edge of the gap (solid line in Fig. 1b-d).”

to

“An analysis of the temperature dependence shows that the two-fermionic character increases by about 120 % at B_c = 6.7 T (when zigzag order is suppressed) compared to B = 0 T (see Supplementary Information, section S3). This can be taken as further evidence for the presence of Majorana fermionic excitations in α -RuCl₃. As the magnetic field increases above 10 T, C becomes gapped and its spectral width narrows down. This opening of the gap is traced by the dashed curve in Fig. 1b.”

(12)

Results

“The excitation M2' (orange line; the notation 'prime' is used to differentiate distinct modes in the high field regime B > B_c) is split off from the M1 peak above 12 T, while the higher-energy M3' excitation (red line) appears at the lower boundary of the gapped continuum above 10-14 T.”

to

“The excitation 2M (orange line) is split off from the M1 peak above 12 T, while the higher-energy 3M feature (red line) appears at the lower boundary of the gapped continuum above 10-14 T.”

(13)

Results

“In consideration of the narrow spectral form and energy of the corresponding excitations observed in our data, we assign the M2' peak to a two-magnon bound state and the M3' peak to either a multimagnon excitation or a van Hove singularity of the gapped continuum.”

to

“In consideration of the narrow spectral form and energy of the corresponding excitations observed in our data, we assign the 2M peak to a two-magnon bound state. The 3M peak could be either the excitation of a three-magnon bound state, or a van Hove singularity of the gapped continuum. Considering the 3M peak sharpens in the high-field regime, the assignment to a three-magnon bound state is more convincing. As the field is lowered to the intermediate phase, the 3M excitation is no longer well-defined.”

(14)

Fig. 2 Caption

“The M1 (blue) and M2 (purple shading) modes at low fields of 0 - 4.3 T correspond to spin wave excitations. The excitation MB (dark red) above the critical field of 6.7 T is assigned to a Majorana bound state.”

to

“The M1 (blue) and M2 (purple shading) modes at low fields of 0 - 4.3 T correspond to spin-wave excitations. The mode (M3) may be a higher-energy branch of magnon excitations. The excitation MB (dark red) above the critical field of 6.7 T is assigned to a Majorana bound state. The shoulder (MB') is either another Majorana bound state or a van-Hove singularity of the Majorana continuum excitations.”

(15)

Results

“In particular, the energy and field ranges of M3' are sensitive to the in-field directions, indicating the presence of non-negligible in-plane anisotropy terms.”

to

“In particular, the energy and field ranges of 3M are sensitive to the in-field directions, indicating the presence of non-negligible in-plane anisotropy terms.”

(16)

Results

“Our high-field Raman data evidence the existence of multimagnon excitations and a gapped continuum that characterizes the spin dynamics of the partially polarized phase.”

to

“Our high-field Raman data evidence the coexistence of multimagnon excitations and a gapped continuum that characterizes the quasiparticle landscape of the partially polarized phase.”

(17)

Results

“Simultaneously, anyonic excitations emerge in the bulk. A detection of these anyonic Majorana bound states will provide an ultimate confirmation of the field-induced intermediate non-Abelian phase.”

to

“Simultaneously, Majorana bound states with possibly anyonic character emerge in the bulk. A detection of these novel excitations will provide a clue to the topological nature of the field-induced intermediate phase.”

(18)

Results

“Compared to the T=5 K high-field data shown in Fig. 1a, the T=2 K data show a new sharp structure at 5 meV (M2) in addition to the one-magnon excitation (M1) and the fractionalized continuum (C).”

to

“Compared to the T=5 K high-field data shown in Fig. 1a, the T=2 K data show new sharp M2 and (M3) structures at 5 meV and 7.5 meV, respectively, in addition to the one-magnon excitation M1 and the fractionalized continuum C. The M2 peak may be tentatively assigned to a two-magnon-like excitation in the singlet sector. Unlike conventional two-magnon scattering, the 2M peak is narrow possibly because a large portion of the two-magnon excitation decays into the continuum.”

(19)

Results

“A new low-energy mode (MB) evolves from the low-field M2 mode with a shoulder structure (M3') and the continuum of Majorana excitations is gapped above 8.1 T.”

to

“A new low-energy MB mode evolves from the low-field M2 mode and a shoulder structure (MB') appears out of the (M3) peak. Apparently, the continuum of Majorana excitations is gapped above 8.1 T.”

(20)

Fig. 3 Caption

“e, Thermally induced binding-unbinding crossover at B=6.7 T and 9.5 T. Grey lines are guides to the eyes.”

to

“e, Thermal evolution of the MB mode energy at B=6.7 T and 9.5 T. Grey lines are guides to the eyes.”

(21)

Results

Added sentence: **“Here, full squares, empty circles, and empty triangles denote B // (100), B // (010), and B // (110), respectively.”**

(22)

Results

“As the magnon corresponds to a condensation of Majorana fermions, the M2-to-MB mode evolution may be interpreted in terms of a condensation-to-confinement crossover where the multimagnon excitations observed at low field gradually evolve into Majorana bound states.”

to

“As the magnon corresponds to a condensation of Majorana fermions, the M2-to-MB mode evolution may be interpreted in terms of a condensation-to-confinement crossover where the magnon excitations observed at low field gradually evolve into Majorana bound states in the singlet sector.”

(23)

Results

“...the low-energy bound state can be naturally created within the gap due to confinement.”

to

“...the low-energy bound state can be created within the gap due to confinement.”

(24)

Results

“Unlike the M2-to-MB mode crossover below B_c , the MB mode is not smoothly linked to the M2' bound state for fields above B_c . Rather, as the field increases, the M2' mode splits from the M1 mode, and both excitations are observed prominently, while a signature of the MB mode remains absent in the data obtained from the high-field setup. This suggests that the MB mode is of different nature than the excitations M1 and M2', and that the parameters temperature, scattering geometry, and magnetic field direction play a crucial role in the creation and observation of Majorana bound states.”

to

“Unlike the M2-to-MB mode crossover below B_c , the MB mode is not smoothly linked to the 2M bound state for fields above B_c . Rather, as the field increases, the 2M mode splits from the M1 mode, and both excitations are observed prominently. This suggests that the high-field phase stabilizes conventional quasiparticles against the fractional excitations. Interestingly, a signature of the MB mode remains absent in our data obtained from the high-field setup, as well as in data from a related recent high-field Raman and THz study [Sahasrabudhe-19], implying that the parameters temperature, scattering geometry, and magnetic field direction are decisive in stabilizing Majorana bound states. It also suggests that we cannot assert a direct relation between the two excitations M2 (at low fields) and 2M (at high fields).”

(25)

Results

“...the small temperature difference of ~ 3 K cannot account for a jump of 1 meV.”

to

“Yet, the small temperature difference of ~ 3 K can hardly account for a jump of 1 meV.”

(26)

Results

“Further support of the MB state interpretation comes from the temperature dependence at two different magnetic fields, 6.7 T and 9.5 T (Fig. 3e).”

to

“Further support of the MB state interpretation comes from the temperature dependence at two different magnetic fields, 6.7 T and 9.5 T (see Fig. 3e and Fig. S9).”

(27)

Results

“For bound states, however, the thermal energy competes with the binding energy [Choi-13], until eventually an unbinding takes place. Therefore, a sudden drop in mode energy occurs at around 15 K (at $B=9.5$ T), setting the binding energy scale to ~ 1.3 meV (see Suppl. Information, section S5 for details). Correspondingly, in a smaller magnetic field of only 6.7 T an unbinding already occurs around $T \sim 10$ K. In contrast, the shoulder of built-up Majorana fermionic excitations gives no clear sign of binding.”

to

“For bound states, however, the thermal energy competes with the binding energy [Choi-13], until eventually a thermally induced unbinding takes place. We can trace the energy of the MB excitation up to temperatures of 9 K (for $B = 6.7$ T) and 13 K (for $B = 9.5$ T). At higher temperatures the MB quasiparticles melt and decay into the higher-energy continuum, and therefore become ill-defined. Based on the peak-energy shift observed in the narrow, limited temperature window, we estimate a lower limit of the binding energy of 0.5 meV. Estimating the binding energy from the gap size yields a slightly higher value of about 1 meV (see Suppl. Information, section S5).”

(28)

Discussion

“This interpretation is supported by the smooth crossover from the multimagnon M2 mode to the bound state MB through B_c (see Figure 3d).”

to

“This interpretation is supported by the smooth crossover from the M2 magnon mode to the bound state MB through B_c (see Figure 3d), as both excitations arise from a spin-conserving scattering process.”

(29)

References

Added Ref. 29, Sahasrabudhe, A. et al. High-Field Quantum Disordered State in α -RuCl₃: Spin Flips, Bound States, and a Multi-Particle Continuum. Preprint at <https://arxiv.org/abs/1908.11617> (2019).

(30)

Acknowledgments

“We acknowledge important discussions with Natalia Perkins.”

to

“We acknowledge important discussions with Natalia Perkins, Yuji Matsuda, and Stephen Nagler.”

(31)

Supplementary S1

“In-plane and out-of-plane Raman scattering experiments at the high magnetic field lab in Grenoble have been carried out in Voigt- and in Faraday geometry, respectively, as sketched in Figs. S1a and S1b.”

to

“Raman scattering experiments with in-plane and out-of-plane magnetic fields have been carried out at the high magnetic field lab in Grenoble in Voigt- and in Faraday geometry, respectively, as sketched in Figs. S1a and S1b.”

(32)

Supplementary S2

Added Fig. S3, Field-induced phonon anomalies.

Added text:

“Here we note that for magnetic fields along the (100) direction the phonon at 20 meV [$E_g(2)$, corresponding to an in-plane displacement of Ru ions] appears to be splitting with increasing fields. To analyze this effect in detail we fit this spectral region with a sum of asymmetric Fano- and symmetric Voigt profiles (see Fig. S3a-c). At low fields the excitation can be mainly described by a single Fano profile (green line), although a faint shoulder exists (blue line). With increasing fields the energies of these modes remain unaffected, but their intensity ratio changes, which results in a prominent growth of the shoulder feature. We assign the occurrence of these two lines to a lifting of the twofold degeneracy of the $E_g(2)$ phonon, pointing towards a minute in-plane lattice distortion. As the energies of these two lines are unaffected by magnetic fields, this distortion is not a field-induced structural transition (see Fig. S3d). With increasing field, a second, broader excitation approaches the 20 meV phonon energetically (dark red line), and around $B = 22$ T locks in with the phonon. This leads to a decrease in phonon lifetime, as evidenced by a decrease in intensity and a broadening in linewidth [see Fig. S2 for $B // (100)$].”

(33)

Supplementary S3

“In order to study the competition of quantum and thermal fluctuations and to analyze the statistics of excitation spectrum C in the intermediate field phase with suppressed LRO we perform experiments as a function of temperature at a fixed magnetic field $B = B_c$.”

to

“In order to study the competition of quantum and thermal fluctuations and to analyze the statistics of excitation spectrum C in the intermediate field phase with suppressed long-range order we perform experiments as a function of temperature at a fixed magnetic field $B = B_c$.”

(34)

Supplementary S3

Added Fig. S5, Decomposition into bosonic- and fermionic excitations.

(35)

Supplementary S3

“Our observations suggest that the Majorana-related excitations at the quantum critical field B_c become more pronounced than the $B=0$ T excitation in spite of the development of the low-energy Majorana bound state MB. This suggests that the field-induced phase is thus closer to a spin liquid than the zero-field phase.”

to

“Our observations suggest that the Majorana-related excitations at the quantum critical field B_c become more pronounced than the $B=0$ T excitation, while developing the low-energy Majorana bound state MB. In Figs. S5a and S5b we plot the as-measured continuum intensity for 0 T and 6.7

T, respectively, together with their bosonic (yellow-shaded background) and two-fermionic (purple-shaded background) contributions. In Fig. S5c we plot the continuum intensity after subtracting the bosonic contributions, with purely two-fermionic contributions remaining. In order to quantify the field-induced change in bosonic-to-fermionic excitations, we determine the ratio of integrated intensities of the two-fermionic (I_{2-F}) and the bosonic (I_B) contributions in the temperature range 0 K - 200 K. We obtain $I_{2-F} : I_B = 0.40$ at $B = 0$ T, and $I_{2-F} : I_B = 0.89$ at $B = 6.7$ T. Thus, the two-fermionic contribution increased by about 120 % upon driving α -RuCl₃ towards quantum criticality, underlining that the field-induced phase is closer to a spin liquid than the zero-field phase.”

(36)

Added Supplementary S5: Polarization dependence

Added Fig. S7

Added text:

“Symmetry considerations can help identify the nature of observed emerging excitations. In Figs. S7a-b we compare spectra measured at base temperature in RL (black line) and LL (red line) polarization, corresponding to the E_g and A_g symmetry channel, respectively. Notably, below as well as above B_c the spectral weight of magnetic excitations is clearly dominant in E_g symmetry and hardly discernible in LL polarization. Only a shallow weak background is detected with an onset energy of around 12 meV, suggesting that the continuum contains a minute contribution of A_g symmetry. In contrast, we do not find any signature of either M1 or M2 in A_g . Note that we also do not observe any leakage of E_g phonons in A_g scattering configuration and vice versa, signaling a very clean experimental polarization. The observed continuum coincides energetically with well-established reports of Majorana fermionic excitations [Sandilands-15, Glamazda-17]. Our very pronounced polarization dependence is in agreement with the theoretically predicted selection rules for the pure Kitaev model [Knolle-14]. Therefore we can confidently assert that the broad scattering signal is dominated by excitations of Majorana fermions rather than magnonic multiparticles. The shallow, finite continuum in LL can be related instead to a deviation from the pure Kitaev model and arising from additional Heisenberg terms in the Hamiltonian. Above B_c , the continuum in LL is even more suppressed, suggesting that the magnetic field freezes out any detrimental impact of the weak perturbation terms. The fact that the intense, narrow mode MB at high fields obeys the same selection rules as the continuum, as well as the same selection rules as the M2 mode, gives evidence for a close relation between these excitations.”

(37)

Supplementary S6

Added Fig. S9: Thermal evolution of the MB mode at two different magnetic fields.

(38)

Supplementary S6

“From an analysis of the T dependence of the peak energy (see Fig. 3e, main text), we estimate the binding energy as $E_B = \omega_{MB} - \Delta$, where $\omega_{MB}=5$ meV and Δ are the energy of MB and the excitation gap of the continuum C, respectively. With a binding-unbinding transition temperature around 15 K we estimate E_B to be around 1.3 meV, which places the gap energy around $\Delta = 6.25$ meV. This estimation is in very good agreement with the fit of the continuum in Fig. 2c, main text (at $B = 9.5$ T). It furthermore implies that the enhanced spectral weight around 8 meV lies above Δ and is part of the field-induced spectral re-distribution of Majorana fermionic excitations, indicative of an opening of the large energy gap due to the sizable Γ -term.”

to

“We can now estimate the binding energy E_B of the MB mode based on its energy $\omega_{MB} \sim 5$ meV and the gap size $\Delta \sim 6$ meV (see Fig. 2c at $B = 9.5$ T) via $E_B = \Delta - \omega_{MB}$. This estimation yields a rough binding energy of 12 K. Considering that above 13 K the MB mode becomes ill-defined, melts and decays with the continuum (see Fig. S9 and Fig. 3e, main text), our estimation proves to be a reasonable one. As the MB mode broadens in linewidth and gradually vanishes with increasing temperature, a narrow shoulder-like feature is observed around 5 meV (see $B = 9.5$ T at $T = 18$ K

and 23 K). We tentatively assign this kink to a van-Hove singularity in the continuum. Note that such a structured continuum is well consistent with calculations for the $K-\Gamma$ model (see solid red lines in Fig. S8).”

REVIEWERS' COMMENTS:

Reviewer #1 (Remarks to the Author):

In the response letter, the authors have well addressed my questions. They have also made several changes in the revised manuscript and improved their discussions. I recommend it to be published on nature communications.

Reviewer #2 (Remarks to the Author):

I am satisfied with the authors' responses. I recommend the publication of the manuscript.